

# Effects of self-selected *versus* motivational music on lower limb muscle strength and affective state in middle-aged adults

Francesca Greco[1,2], Luca Rotundo[3], Elisa Grazioli[1,2], Attilio Parisi[1], Attilio Carraro[4], Carolina Muscoli[5], Antonio Paoli[3], Giuseppe Marcolin[3] and Gian Pietro Emerenziani[2]

[1] Department of Human Movement Sciences and Health, University of Rome "Foro Italico", Rome, Italy
[2] Department of Experimental and Clinical Medicine, University "Magna Græcia" of Catanzaro, Catanzaro, Italy
[3] Department of Biomedical Sciences, University of Padova, Padova, Italy
[4] Faculty of Education, Free University of Bozen, Bozen, Italy
[5] IRC-FSH Department of Health Sciences, University "Magna Græcia" of Catanzaro, Catanzaro, Italy

## ABSTRACT

**Background:** Strength training plays a crucial role in promoting healthy ageing and music might affect how individuals perform and perceive strength exercises. This study aimed to investigate the effects of self-selected music (SSM) on muscle strength and affective states during maximal isometric contractions on a customized leg extension.

**Methods:** Twenty-six healthy middle-aged males (50.8 ± 8.4 years) performed maximal and endurance isometric strength tests under three different conditions: SSM, motivational music (MM), and control condition (CC). Peak force and Rate of Force Development (RFD) were assessed during the maximal isometric strength test. The isometric endurance test evaluated the mean force and a fatigue index. Moreover, Felt Arousal Scale (FAS) was administered before the strength protocol, whereas the Rate of Perceived Exertion (RPE) and Feeling Scale (FS) at the end of it.

**Results:** Mean force was significantly higher in the SSM (507.3 ± 132.2 N) than MM (476.3 ± 122.4 N, $p < 0.01$) and CC (484.6 ± 119.2 N, $p = 0.03$). FAS was significantly higher in the SSM (4.0 [1.3] than MM (3.0 [2.3], $p < 0.01$) and CC (3.0 [1.3], $p < 0.01$) conditions. FS was significantly higher in the SSM (4.0 [2.0] than MM (3.0 [1.3], $p < 0.01$) and CC (3.0 [1.3], $p < 0.01$) conditions. No significant differences were found for peak force, RFD, fatigue index, and RPE.

**Conclusions:** Listening to SSM seems to influence isometric endurance strength performance in middle-aged adults positively. Moreover, listening to SSM might improve individuals' affective states without affecting the level of perceived exertion.

# INTRODUCTION

Regular practice of physical activity (PA) is an essential addition to healthy living and its positive effects are recognized at all ages. Physical activity represents a protective factor

Corresponding author
Gian Pietro Emerenziani,
emerenziani@unicz.it

against several diseases (*e.g.*, cardiovascular diseases, diabetes, chronic respiratory diseases, and some types of cancers), counteracting all the harmful-related effects of physical inactivity (*Warburton, Nicol & Bredin, 2006*; *Taylor, 2014*; *Pedersen & Saltin, 2015*). The multi-systemic and unique health-related benefits of muscle strengthening exercises are broadly acknowledged (*Bennie, Shakespear-Druery & De Cocker, 2020*; *Maestroni et al., 2020*). Indeed, resistance training alone lowers 21% of all-cause mortality and increases cardiometabolic health, muscle mass, musculotendinous integrity, force, and power while decreasing the onset of developing physical functional limitations (*Bennie, Shakespear-Druery & De Cocker, 2020*; *Saeidifard et al., 2019*; *Garber et al., 2011*). Moreover, lower limb muscle strength, muscle endurance and muscle power represent a predictor of mobility and functional limitations related to everyday functional tasks (*Mitchell et al., 2012*; *Maffiuletti et al., 2016*). The life-course paradigm encourages people to adopt healthier lifestyles earlier in life. Healthy aging over the life course is also recognized as a fundamental strategy for preventing of several disabling indicators (*e.g.*, sarcopenia, risk of falls, disabilities) (*World Health Organization, 2012*; *Tieland, Trouwborst & Clark, 2018*).

Therefore, strength training plays a crucial role in maintaining and improving muscle integrity (*Bennie, Shakespear-Druery & De Cocker, 2020*; *Fragala et al., 2019*; *Mcleod, Stokes & Phillips, 2019*) also in middle-aged adults. Even though scientific evidence supports the positive effects of strength training on health and well-being, over 80% of adults do not reach the muscle-strengthening guidelines (≥ 2 or more days per week) (*Bennie, Shakespear-Druery & De Cocker, 2020*). This negative evidence led to a worldwide public health priority carrying out strategies to encourage strength training. Furthermore, this scenario becomes worse as physical inactivity levels increase with age (*Taylor, 2014*).

Listening to music might positively influence the adherence and pleasure of undertaking PA, improving the efficiency of the performance (*Terry et al., 2020*). Indeed, it has been shown that music positively influences affective valence during PA, resulting in enhanced exercise adherence (*Clark, Baker & Taylor, 2016*). Music could act as ergogenic aid enhancing psychological (*e.g.*, affective states), psychophysical (*i.e.*, perception of one's physical state), and physiological (*e.g.*, heart rate) responses (*Terry et al., 2020*; *Karageorghis & Priest, 2012*). It is generally accepted that music reduces the level of perceived exertion in low-to-moderate PA and, if well-selected, enhances exercise-related affect status acting as a mild ergogenic aid (*Karageorghis, 2020*). Scientific literature has mainly focused on the effects of motivational (music tempo > 120 bpm) and self-selected music on muscular fitness (*Crust, 2004*; *Biagini et al., 2012*; *Silva et al., 2021*; *van den Elzen et al., 2019*). Specifically, self-selected motivational music may positively affect muscle strength performance while exercising (*Crust, 2004*; *Bartolomei, Di Michele & Merni, 2015*; *Ballmann et al., 2021*; *Silva et al., 2021*). *Feiss et al. (2021)* showed that muscle strength performance was not affected by different music tempos (120 *vs.* 90 bpm). Moreover, listening to self-selected music improves muscle strength (*van den Elzen et al., 2019*; *Biagini et al., 2012*). Findings showed that both motivational and self-selected music might improve endurance strength, grip strength, explosive strength, and motivation (*Crust, 2004*; *Biagini et al., 2012*; *Bartolomei, Di Michele & Merni, 2015*; *Ballmann et al., 2021*; *van*

*den Elzen et al., 2019*) while decreasing the level of perceived exertion (*Biagini et al., 2012*; *Silva et al., 2021*). Indeed, literature regarding the effects of music during PA showed that human movement and perception are influenced by 120 bpm music tempo (*i.e.*, intrinsic factor). Noteworthily, 120 bpm is twice the resting heart rate of healthy adults and the preferred walking step frequency in humans (*Karageorghis, 2020*; *Terry et al., 2020*). Musical preferences have been recognized as a moderator factor that might influence the relationship between a musical stimulus and the individuals' responses to it. It has been assessed that familiarity with a melody might positively influence emotional responses, thereby increasing exercise performance (*Karageorghis, 2020*; *Terry et al., 2020*; *Greco et al., 2022*). However, it is not yet clear whether the main effect on muscle strength is given by the music characteristics (*i.e.*, tempo > 120 bpm) or the affective responses consequent to listening to self-selected music.

Since strength training assumes a crucial role in a successful aging process, it would be interesting to understand whether listening to preferred music might increase strength performance in middle-aged adults. Moreover, studies on the effects of different types of music administration (*i.e.*, self-selected *vs.* experimenter-selected music) on lower limb isometric exercises have not been carried out yet. Therefore, the primary aim of the study was to investigate the effects of self-selected (*i.e.*, selected by the participants) and motivational music (*i.e.*, selected by the researchers) on lower limb maximal and endurance isometric strength in healthy middle-aged adults. Also, we investigated if different types of music could affect the level of perceived exertion and affective responses.

## MATERIALS AND METHODS

### Study design

The study adopted a randomized repeated measure design. Participants attended the Physical Exercise and Sports Science laboratory five times at the University of "Magna Graecia" of Catanzaro. Two familiarization sessions were employed to assess participants' characteristics and accustom them to the experimental protocol. Then, each participant underwent in three separate days a lower limb isometric leg extension protocol in a randomized order (www.random.org) under three conditions: self-selected music (SSM), motivational music (MM), and control condition (CC). Each test condition was separated by 72 h to avoid fatigue and muscle soreness. Tests were performed with the dominant leg. Written informed consent was obtained and research procedures complied with the Declaration of Helsinki. The study was approved by the local institutional review board (department of experimental and clinical medicine, University of Magna Graecia) and by the ethics committee of the Calabria region (n. 122).

### Participants

Twenty-nine male participants were enrolled in the current study. Twenty-six participants (Mean ± Standard Deviation (SD); Age = 50.8 ± 8.4 years; BMI = 25.8 ± 3.4 kg/m$^2$) completed the whole experimental procedure, as three participants dropped out. Inclusion criteria were not being engaged in regular resistance training, aged 40–64 years old. Exclusion criteria were any contraindication to PA according to the physical activity

**Table 1 Participants' characteristics.** Data are presented as Mean ± SD.

| | |
|---|---|
| Age (years) | 50.8 ± 8.4 |
| Height (cm) | 174.9 ± 7.0 |
| Body Mass (kg) | 79.0 ± 9.8 |
| BMI (kg/m$^2$) | 25.8 ± 3.4 |
| PAL (METs-min/week) | 1528.5 ± 1193.3 |

**Note:**
 BMI, Body Mass Index; PAL, Physical Activity Level.

readiness questionnaire (PAR-Q+) (*Warburton et al., 2019*), hearing impairments, music as unpleasant stimuli, or individuals that had no natural preference or were not able to name a song by themselves, any impairment affecting lower limbs functionality, cognitive impairments. Participants' characteristics involved in the study are shown in Table 1.

## Experimental procedure

### Familiarization sessions

Demographic information (*i.e.*, age, gender, and date of birth), the PAR-Q+ (*Warburton et al., 2019*), and the global physical activity questionnaire (G-PAQ) (*Armstrong & Bull, 2006*) were assessed in the first access to the laboratory. The PAR-Q+ was used to ensure that participants did not require further advice before participating in PA. At the same time, the G-PAQ gave information on PA participation in three domains (activity at work, travel to and from places, and recreational activities). Moreover, researchers asked the participants to provide a playlist of ten preferred songs able to increase their motivation during PA. To avoid the Hawthorne effect, all participants were encouraged to choose their playlist without restrictions (*e.g.*, genre, tempo, language) because they would not have been subjected to any judgment by the researcher. The playlist was given to the researcher before the first test session. After completing these forms, height (to the nearest 0.1 cm) and body mass (to the nearest 0.1 kg) were measured by using a stadiometer with a weighting station (SECA, Intermed S.r.l., Milano, IT). Then, participants familiarized themselves with the customized leg extension by performing isometric sub-maximal and endurance tests.

### Music administration

In the SSM condition, participants listened to their playlist in which songs were played randomly. For the MM condition, researchers selected the songs from the top Hits Italia with the Spotify application (Spotify Technology S.A., Stockholm, Sweden) to guarantee familiarity and cultural appropriateness (*i.e.*, currently the most played songs on the radio). To ensure that songs fell within the characteristics of motivational music, only musical tracks above 120 bpm were selected (*Karageorghis, 2020*). Thus, the mean tempo of the MM was 128.8 ± 4.0 bpm and the style of the songs included Pop, Hip-hop, and Electronic dance music. All participants heard the MM playlist in the same order. Music was administered through the different test conditions (SSM, MM) using the same headphones (Beats studio3 Wireless; Apple Inc., Culver City, CA, USA) *via* noise-isolation to avoid
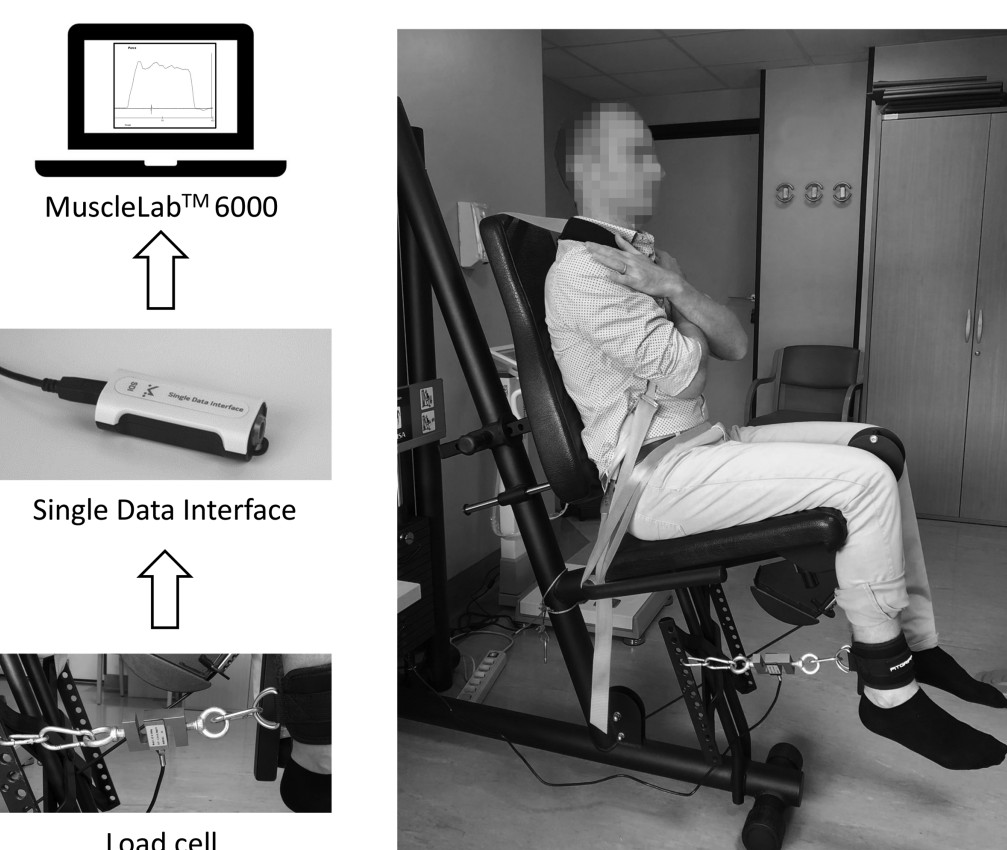

**Figure 1 Force data acquisition setup.**

confounding effects. The level of the volume was adjusted to 75 dB. Music was played through a mobile phone (iPhone 8, Apple Inc., Cupertino, CA, USA). In the CC condition, the participants wore headphones, but no music was played.

### Test sessions

Testing sessions were performed in a time window from 10:00 am to 06:00 pm. Each participant undertook all the test sessions at the same clock time (*i.e.*, time-matched). In each session, participants wore headphones from the beginning to the end of the protocol.

The warm-up consisted of 5 min on the cycle ergometer (Ergoselect, ergoline GmbH, Bitz, Germany) cycling at 50 revolutions per minute (rpm) at a self-selected load (ranging from 25 to 75 watts) followed by ten dynamic half-squats. Afterward, participants performed the isometric leg extension protocol on a modified leg extension machine (Nextline Leg extension; Visa Sport, Marcellinara, Italy) with the dominant lower limb. The force data acquisition setup is reported in Fig. 1. Briefly, the dominant limb was blocked on a frame with a knee angle of 90 degrees with adjustable belts. The belt on the ankle was collected through a strap (Fitgriff; GmbH, Bergheim, Austria) to a load cell (MuscleLab™ 6000; MuscleLab, Porsgrunn, Norway), enabling the measurement of the force expressed. Moreover, the trunk-thigh angle was set at 90 degrees and a series of straps

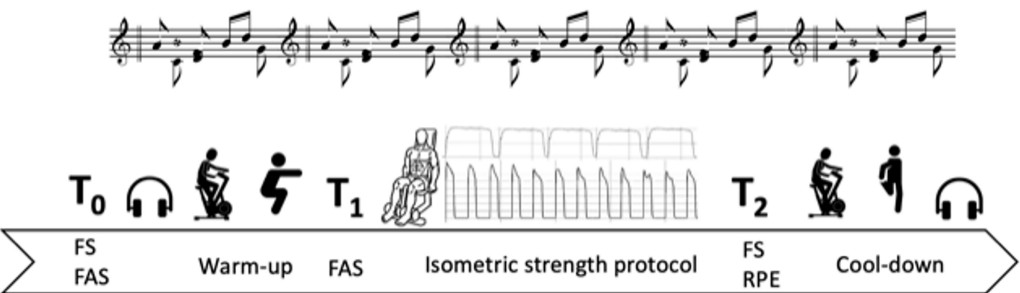

**Figure 2 Graphical representation of the test sessions.** FS, Feeling Scale; FAS, Felt Arousal Scale; RPE, Rate of Perceived Exertion.

across the shoulders and pelvis minimized body movements during the tests. Participants, with arms crossed, were instructed to perform the maximal isometric strength test pushing as hard and fast as possible. The maximal isometric test consisted of five maximal isometric contractions held for 3 s interspersed by a rest of 60 s. Since headphones were worn, verbal encouragement was not given, and a visual countdown (timer) was used to give the participant the exercise timing. Then, after 3 min of rest, participants performed the endurance strength exercise, consisting of twelve maximal isometric contractions held for 3 s interspersed with 5-s rest between each repetition. The protocol ended with 5-min on a cycle ergometer plus lower limb stretching exercises.

The Felt Arousal Scale (FAS) (*Svebak & Murgatroyd, 1985*), the Feeling Scale (FS) (*Hardy & Rejeski, 1989*), and the RPE scale (CR-10) (*Borg, 1998*) were administered at specific time points of the strength tests. Specifically, FAS, FS, and RPE were administered before the warm-up of each test session to avoid differences among test conditions ($FAS_{T0}$, $FS_{T0}$, $RPE_{T0}$). FAS was then administered before the beginning of the isometric leg extension protocol ($FAS_{T1}$), whereas FS, together with RPE, at the end of the isometric leg extension protocol ($FS_{T2}$ and $RPE_{T2}$). A graphical representation of the test session is presented in Fig. 2.

## Data analysis

The MuscleLab^TM 6000 system was used to assess muscle strength variables. The load cell recorded the force output at 200 Hz using a single data interface (Fig. 1) and its dedicated software (MuscleLab v. 10.213.98.5188, MuscleLab^TM 6000 Ergotest Innovation; MuscleLab, Porsgrunn, Norway). The mean of the best three measurements was considered to analyze both peak force and RFD values. Specifically, we applied a 15-Newton threshold to avoid baseline noise. Then, RFD was calculated over a time window of 50 milliseconds, starting from the force signal onset.

Endurance was assessed through the mean force parameter, computing the mean forces of the twelve contractions for each participant. Moreover, a fatigue index was calculated as follows (*White et al., 2013*): [(maximal force at 1^st repetition−maximal force at 12^th repetition)/maximal force at 1^st repetition] * 100).

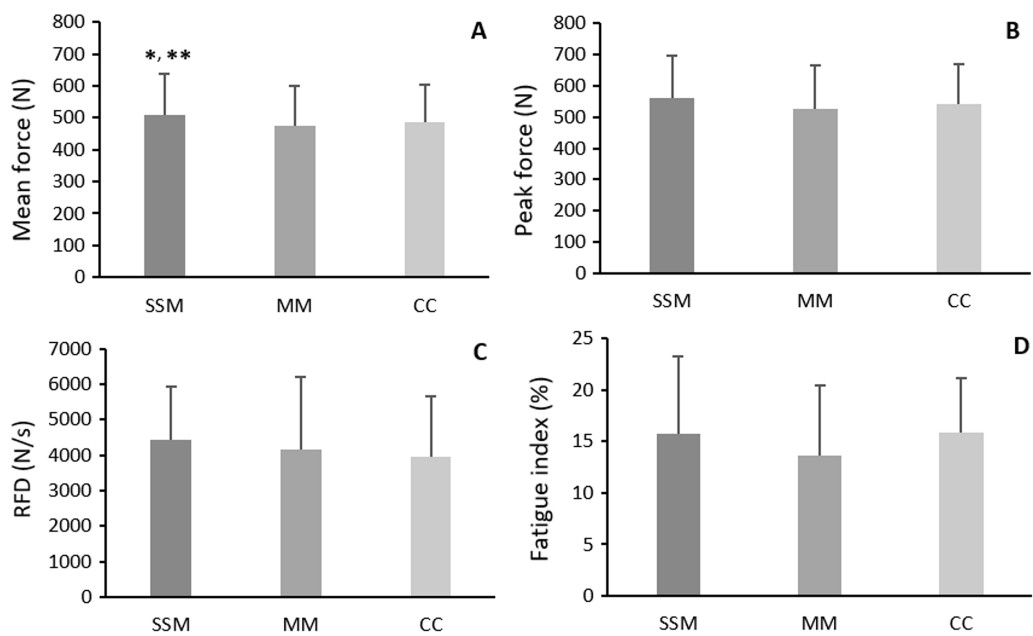

**Figure 3 Results of muscle strength.** (A) Mean force during endurance test; (B) peak force during maximal strength exercise; (C) rate of Force Development during maximal strength exercise; (D) fatigue index value during the endurance test. Data are presented as Mean ± SD. SSM, Self-Selected Music; MM, Motivational Music; CC, Control Condition; *$p$ = 0.03 *vs*. CC; **$p$ < 0.01 *vs*. MM.

## Statistical analysis

*A priori* power analysis calculation (G*Power 3.1.9.2 software) showed that a sample size of 26 participants and a medium effect size of 0.25 would provide a power of 0.8. Before further analysis, the normal distribution of the dependent variables was tested by applying the Shapiro-Wilk test. This test showed that physiological variables (*i.e.*, peak force, RFD, mean endurance force, and fatigue index value) were normally distributed, while FAS, FS, and RPE had skewed distributions. Therefore, one-way ANOVA for repeated measures on condition factor (SSM, MM, CC) was conducted to detect the effects of music on peak force, RFD, mean endurance force, and fatigue index values. The assumption of sphericity for each ANOVA model was checked with the Mauchly's sphericity test. When significant differences were detected, Bonferroni *post hoc* analysis was run. FAS, FS, and RPE values were analyzed with the nonparametric Friedman's test followed, whenever significant, by the Wilcoxon test for matched pairs as *post hoc* analysis. The level of significance was set at $p$ < 0.05. Statistical analysis was implemented with IBM®SPSS statistics software version 23.0 (SPSS Inc., Chicago, IL, USA).

## RESULTS

There was no overlap between SSM and MM songs. Songs selected for the SSM condition were very different among participants, with an overall mean tempo equal to 107.4 ± 42.4 bpm. The list of the whole songs chosen by participants is reported in Data S1. Results of muscle strength are reported in Fig. 3. A significant main effect of condition ($F_{2,50}$ = 7.534,

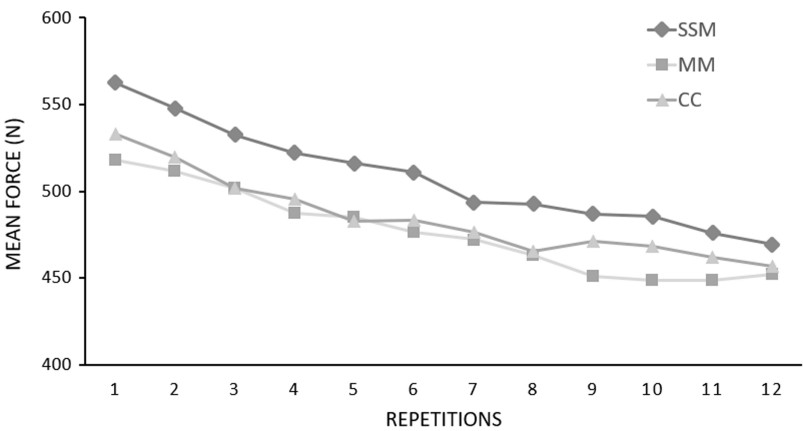

**Figure 4 The peak forces expressed during each of the 12 repetitions and averaged over the 26 participants during the endurance test.** SSM, Self-Selected Music; MM, Motivational Music; CC, Control Condition.

**Table 2 Results of FAS, FS and RPE. Data are presented as Median (IQR).**

**Test conditions**

| Variable | SSM | | MM | | CC | |
|---|---|---|---|---|---|---|
| $FAS_{T0}$ | 3.0 | (2.0) | 2.0 | (3.0) | 3.0 | (2.3) |
| $FS_{T0}$ | 3.0 | (2.0) | 3.0 | (2.0) | 3.0 | (4.0) |
| $RPE_{T0}$ | 0.0 | (0.0) | 0.0 | (0.0) | 0.0 | (0.0) |
| $FAS_{T1}$ | $4.0^{\alpha,\beta}$ | (1.3) | 3.0 | (2.3) | 3.0 | (1.3) |
| $FS_{T2}$ | $4.0^{\alpha,\beta}$ | (2.0) | 3.0 | (1.3) | 3.0 | (1.3) |
| $RPE_{T2}$ | 3.0 | (3.0) | 3.0 | (3.0) | 3.0 | (3.3) |

**Notes:**
SSM, Self-Selected Music; MM, Motivational Music; CC, Control Condition; FAS, Felt Arousal Scale; FS, Feeling Scale; RPE, Rate of Perceived Exertion.
$^{\alpha}p < 0.01$ *vs.* MM; $^{\beta}p < 0.01$ *vs.* CC.

$p < 0.01$, $\eta^2 = 0.232$) on mean force during the endurance test was observed. *Post hoc* analysis highlighted a higher value in the SSM ($507.3 \pm 132.2$ N) than MM ($476.3 \pm 122.4$ N, $p < 0.01$) and CC ($484.6 \pm 119.2$ N, $p = 0.03$) conditions (Fig. 3A). The peak forces expressed during each of the 12 repetitions and averaged over the 26 participants during the endurance test are presented in Fig. 4. No significant differences among conditions for peak force ($F_{2,50} = 3.047$, $p = 0.06$, $\eta^2 = 0.109$) (Fig. 3B), RFD ($F_{2,50} = 2.335$, $p = 0.107$, $\eta^2 = 0.085$) (Fig. 3C) and fatigue index value ($F_{2,50} = 1.553$, $p = 0.222$, $\eta^2 = 0.058$) (Fig. 3D) were found.

Results of psychological and psychophysical variables are reported in Table 2. No significant differences among conditions for $FAS_{T0}$, $FS_{T0}$, and $RPE_{T0}$ were found (Table 2). A significant main effect of condition ($\chi^2(2) = 19.303$, $p < 0.01$) on $FAS_{T1}$ was observed. Indeed, $FAS_{T1}$ was significantly higher in the SSM than in MM ($p < 0.01$) and CC conditions ($p < 0.01$) (Table 2). Moreover, a significant main effect of condition ($\chi^2(2) = 22.594$, $p < 0.01$) on $FS_{T2}$ was observed. $FS_{T2}$ was significantly higher in the SSM

than MM ($p < 0.01$) and CC conditions ($p < 0.01$) (Table 2). No significant differences among conditions for $RPE_{T2}$ ($\chi^2(2) = 1.506$, $p = 0.471$) was found (Table 2).

## DISCUSSION

This study investigated the influence of music on lower limb muscle strength in healthy middle-aged adults for the first time. Our findings revealed a positive effect of SSM on the mean force in the endurance test compared to MM and CC conditions. Conversely, SSM did not affect peak force, RFD, or fatigue index. The psychophysical and psychological analysis highlighted the positive effect of SSM on affective states (FAS and FS) during the isometric leg extension protocol with no changes in the RPE scale.

Although the absence of differences in the fatigue index value and RPE among conditions, participants produced the highest mean force while listening to a self-selected playlist, which may lead to speculate that SSM acted as an ergogenic aid. Our results align with previous investigations despite the different study designs adopted (type of exercise, music administration, and population involved) (*Crust, 2004*; *Bartolomei, Di Michele & Merni, 2015*; *Silva et al., 2021*). Particularly, *Crust (2004)* and *Bartolomei, Di Michele & Merni (2015)* revealed a greater endurance performance after listening to SSM than in a non-music condition. *Silva et al. (2021)* showed that strength-endurance performance increased compared to non-preferred and no-music conditions when listening to the preferred music. Conversely, *Biagini et al. (2012)* indicated that SSM did not enhance bench-press endurance strength performance. As suggested by the authors, it might be that participants' music selection was not stimulative enough to overcome the physical demands of a prolonged strength exercise (*Biagini et al., 2012*). Nonetheless, our results showed that self-selection of music would be an appropriate strategy to improve strength-endurance performance in middle-aged adults. This positive effect might be due to the emotional and hormonal responses triggered while listening to preferred music (*Wilkins et al., 2014*; *Greco et al., 2021*).

Concerning the peak force, our findings supported the literature that showed no ergogenic effects of SSM (*Bartolomei, Di Michele & Merni, 2015*; *Köse, 2018*). However, other authors (*Silva et al., 2021*; *van den Elzen et al., 2019*) obtained different results. Discrepancies might be reconducted to the heterogeneity in the study designs above all considering the type of the exercise (*i.e.*, handgrip test (*Silva et al., 2021*; *van den Elzen et al., 2019*) *vs.* bench press (*Bartolomei, Di Michele & Merni, 2015*; *Köse, 2018*). We can speculate that a maximal voluntary isometric contraction requires maximum physical and mental demand, leading the central nervous system to overcome exercise stress. Consequently, the mind could not focus sufficiently on music, reducing its positive effects.

The RFD results that emerged in the present study contrast with those of *Biagini et al. (2012)*. This result may depend on the different exercises performed (Single leg isometric strength *vs.* concentric squat jump). Since RFD was calculated in the first 50 ms of the muscle contraction that is affected by neuronal components (*i.e.*, motor unit discharge rate) (*Maffiuletti et al., 2016*), we can speculate that SSM may be ineffective in fast and
highly demanding strength tests. Given the paucity of evidence on this neurophysiological aspect, future specific studies are needed to evaluate the effect of different music on RFD.

Regarding the psychophysical response, results showed that SSM did not affect RPE, but participants exerted greater mean force in SSM than in MM and CC conditions. Our findings align with previous investigations that showed no differences in RPE after performing handgrip, wall sit, and plank exercises (*Feiss et al., 2021*; *Bigliassi et al., 2018*) and in contrast with the study of *Silva et al. (2021)*. It is worth noting that fatigue index value also did not change among conditions. Therefore, during SSM, participants had a better endurance-strength performance despite showing the same muscular tiredness among conditions. These results may depend on the effects of SSM on physiological pathways that we did not assess (*e.g.*, nervous system activation) (*Wilkins et al., 2014*; *Bigliassi et al., 2018*). Therefore, future studies using biofeedback systems are needed to understand better how SSM may increase isometric endurance performance. The low RPE value might be justified by the nature of the isometric strength protocol. Indeed, muscle strength exercise consisted of short contractions performed with a single limb for a relatively short time.

Results on the affective state (FS) and activation (FAS) showed that participants felt better and more "worked up" after listening to their favorite songs. Our results highlighted a more significant effect of the selection of the music rather than its intrinsic tempo characteristics (*i.e.*, bpm) on the affective states. Indeed, SSM mean tempo was $107.4 \pm 42.4$ bpm. This might be justified by the emotional response provoked by the preferred music. Regardless of the type, music preference may have additional effects on exercise performance. Indeed, listening to a favorite song affects functional connectivity in regions involved in self-referential thought and memory encoding (*Wilkins et al., 2014*). Compared to our study, the different results obtained by *Bigliassi et al. (2018)* and *Feiss et al. (2021)* might be related to the fact that participants did not select their preferred music. Thus, we may speculate that improved psychological status (FS and FAS) rather than a reduced RPE value led to the highest mean force during an isometric leg extension protocol.

Given the importance of practicing muscle-strengthening exercises during life span, our results highlighted the positive role of listening to SSM to improve the productivity of a single bout of strength training. Indeed, an improved affective state during SSM might maximize health-related benefits and positively influence exercise adherence (*Terry et al., 2020*) to counteract age-related consequences.

We are aware of some limitations of the present study. First, we did not analyze physiological and endocrinological parameters such as the neuromuscular activity and the cortisol levels, which could have contributed to elucidating our findings better. Second, we investigated the muscle strength responses during an isometric exercise and only in middle-aged males. Thus, our results cannot be generalized to all strength-exercises modalities and to different age groups. Moreover, seeing that differences in fatigability during isometric and dynamic exercise have been previously reported between males and females (*Hunter, 2014*, *2016*, *2018*), we cannot generalize our results to females individuals.

## CONCLUSIONS

Listening to self-selected music might increase lower limb isometric strength endurance but not maximal strength performance in healthy middle-aged adults. Moreover, listening to a preferred playlist might improve the affective state with no changes in the level of the perceived exertion. Future studies are needed to assess the possibility of self-selected music to enhance adherence to strength training protocol in middle-aged adults.

### Funding

This work was supported by the Italian Ministry of Education and University (No. 2017FJSM9S). The funders had no role in study design, data collection and analysis, decision to publish, or preparation of the manuscript.

### Grant Disclosures

The following grant information was disclosed by the authors:
Italian Ministry of Education and University: 2017FJSM9S.

### Competing Interests

Gian Pietro Emerenziani is an Academic Editor for PeerJ.

### Author Contributions

- Francesca Greco conceived and designed the experiments, performed the experiments, analyzed the data, prepared figures and/or tables, and approved the final draft.
- Luca Rotundo conceived and designed the experiments, performed the experiments, prepared figures and/or tables, and approved the final draft.
- Elisa Grazioli performed the experiments, prepared figures and/or tables, and approved the final draft.
- Attilio Parisi analyzed the data, authored or reviewed drafts of the article, and approved the final draft.
- Attilio Carraro analyzed the data, authored or reviewed drafts of the article, and approved the final draft.
- Carolina Muscoli analyzed the data, authored or reviewed drafts of the article, and approved the final draft.
- Antonio Paoli analyzed the data, authored or reviewed drafts of the article, and approved the final draft.
- Giuseppe Marcolin conceived and designed the experiments, performed the experiments, authored or reviewed drafts of the article, and approved the final draft.
- Gian Pietro Emerenziani conceived and designed the experiments, performed the experiments, authored or reviewed drafts of the article, and approved the final draft.

## Human Ethics

The following information was supplied relating to ethical approvals (*i.e.*, approving body and any reference numbers):

The study was approved by the local institutional review board (department of experimental and clinical medicine, University of Magna Graecia) and by the ethics committee of the Calabria region.

## Data Availability

The raw measurements are available in the Supplemental File.

## Supplemental Information

Supplemental information for this article can be found online at http://dx.doi.org/10.7717/peerj.13795#supplemental-information.

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

publications/m/item/report-of-the-sixty-second-session-of-the-who-regional-committee-for-
europe.