# Peer review of "Effects of self-selected versus motivational music on lower limb muscle strength and affective state in middle-aged adults"

_PeerJ, doi:10.7717/peerj.13795_

## Round 0.1 · original submission · Major Revisions

Both reviewers found merit in your manuscript but recommended suggestions for improvement. Please consider each reviewer's comments when revising.

Reviewer 1 ·

Basic reporting

Basic Reporting

Overall, the article is well-written, with clear and unambiguous language. The research objective is clear: the authors wished to test the effect of self-selected music versus motivational music on isometric leg strength and endurance as well as affective state in middle-aged adults.

Introduction
The introduction is well-written, though one key change that would help strengthen it is a greater focus on previous work evaluating the effects of music on exercise performance and potentially adherence. Much of this work is referenced in the paper’s discussion, without a great amount of detail in the introduction. Given the study purpose, the introduction could benefit from more discussion of how self-selected music, per se, may confer greater/different performance benefits than motivational music. For example, the authors may expand on the statement and references in lines 95-97 to help lend support the idea that self-selected music may confer some different effect on performance than motivational music. The reviewer acknowledges that the cited references do not directly compare motivational (>120 bpm) and self-selected music, however, findings of improved performance with preferred music over other music from Silva et al. (2021) and van de Elzen et al., (2019) would still be beneficial to highlight and might help reinforce the rationale for comparing self-selected and motivational music as defined in the current study.
Moreover, some discussion of conceptual frameworks of how music may affect exercise performance (Terry et al., 2020) might be warranted. Within this meta-analytic review by Terry et al. that the authors have referenced, Terry et al. describe that there have been several previously published conceptual models (including their own) that propose mechanisms for music’s effects on exercise performance. The authors of the current study state that mechanisms are still being elucidated, (line 98) and it would be helpful to the reader to understand current ideas about such mechanisms. Discussing these ideas may provoke thought about how a difference in performance might result from listening to self-selected music versus motivational music. For example, could self-selecting music differentially enhance affective responses through something like cultural impact (Terry et. al 2020) which motivational music may not? If space or page limitations are currently a concern, some of the discussion of the benefits of incorporating resistance exercise throughout the lifespan might be reduced considering that it is not central to the research question being tested. Perhaps readers could be referred to some comprehensive outside sources for this content.
Please include some discussion of how self-selected music per se might be hypothesized to provide an additional benefit over motivational music to strengthen the rationale for the research question.

Experimental design

Experimental Design

The authors have clearly stated their research question, which is relevant and meaningful considering improved affective state and performance may be beneficial for exercise adherence in the long-term.

The experimental design is adequate to answer the question of whether self-selected music and motivational music may confer different benefits for performance and affective state. There are some clarifications to the Materials & Methods section that would improve reproducibility of the study, and there are also some potential limitations that may be important to acknowledge.

Line 124 – The authors outline “inclusion” criteria, although for example, contraindications to physical activity (PAR-Q+) seem like they would be used to exclude participants. Please clarify whether these are inclusion or exclusion criteria.

Line 127 – It is stated that “music as an annoyance factor” is a criteria for exclusion. How is this assessed? Does this mean that people who do not wish to listen to music while they exercise are excluded? If this is the case, this should be considered a limitation and it should be discussed that this could considerably bias results.

Lines 145-150 – MM music was selected by choosing songs from “Top Hits Italia” that had a tempo above 120 bpm. Is it possible that there could have been considerable overlap between SSM and MM, where some of the tracks were the same? If so, how was this handled?

Line 159 – Were lab visits time-matched within this 10:00 am to 6:00 pm window such that participants returned at the same time of day each visit (e.g. 12:00 pm)? Or were participants free to choose any time to report to the lab for each visit, given it was between 10:00 and 6:00? Please clarify.

Validity of the findings

3) Validity of the Findings
Overall, the authors have clearly reported their statistical analyses and findings, though some minor clarifications would be helpful for improving this.

Line 189 – RFD was calculated using the 0-50ms window. More information is needed to determine how this window was detected. There are many possible ways of doing this (Maffiuletti et al., 2016). Was visual detection of contraction onset used? A statistical method to see when force deviated beyond baseline noise? Please clarify.

Line 200 – One-way repeated measures ANOVA was used to compare normally distributed physiological variables across conditions. Was the assumption of sphericity met for each ANOVA model? If not, was any correction (e.g., Greenhouse-Geisser or Huynh – Feldt) used to account for positive bias in Type I error rate? It doesn’t seem like it would change any outcomes of statistical inference for these variables when looking at the p-values, but transparency would strengthen the reporting.

Line 260 – A reference would be helpful to strengthen this statement. There is a thorough discussion about this in the aforementioned paper by Maffiuletti et al. (2016).

Lines 291-295 – Some discussion of a few other limitations that affect generalizability of the findings is warranted. First, there are distinct sex differences in fatigability during isometric and dynamic exercise (for thorough review, see Hunter 2014, 2016, 2018). Considering that the authors presumably wished to generalize these findings to all middle-aged adults based on the title as well as discussion (e.g. lines 226-227 & 298-299), the limitation to a sample consisting of only middle-aged males must be acknowledged. Please provide a statement about this limitation to generalizability.
Second, there is also the possibility that there were “Hawthorne” effects that may have confounded the self-selected music choices. Terry et al. (2020) provide some discussion of this in their review. For example, participants may have selected music other that what their true self-selected motivational music would be because they knew researchers would have access to and evaluate their choices. Perhaps participants would have chosen differently if researchers did not ask for the 10 songs, but instead participants selected the songs themselves knowing the researchers would be blinded to their choices. Please provide some discussion about this as a possible limitation, or perhaps delimit in some way if this was somehow accounted for in the study design.

References

Hunter, S. K. (2014). Sex differences in human fatigability: mechanisms and insight to physiological responses. Acta Physiol (Oxf), 210(4), 768-789. https://doi.org/10.1111/apha.12234
Hunter, S. K. (2016). The Relevance of Sex Differences in Performance Fatigability. Med Sci Sports Exerc, 48(11), 2247-2256. https://doi.org/10.1249/MSS.0000000000000928
Hunter, S. K. (2018). Performance Fatigability: Mechanisms and Task Specificity. Cold Spring Harb Perspect Med, 8(7). https://doi.org/10.1101/cshperspect.a029728
Maffiuletti, N. A., Aagaard, P., Blazevich, A. J., Folland, J., Tillin, N., & Duchateau, J. (2016). Rate of force development: physiological and methodological considerations. Eur J Appl Physiol, 116(6), 1091-1116. https://doi.org/10.1007/s00421-016-3346-6

Additional comments

This manuscript titled “Effects of self-selected versus motivational music on lower limb muscle strength and affective state in middle-aged adults” is well-written, and the authors have clearly stated their research question and outlined their methodology for answering the question. Overall, the manuscript is clear and informative, and it can be strengthened with some changes and additions.

Reviewer 2 ·

Basic reporting

In general, I found the article well written. The theme is interesting. However, I think that the introduction could be more directed to the research problem to be investigated.
- The first paragraph is too broad and beyond the scope of the article (e.g., lines 52 to 57).
- Lines 66 to 70 focus on older adults, but the paper is not on this population.
- Lines 73 to 77 describe adherence to strength training, a problem that was also not investigated by the present study.
- It was also unclear what was the main gap to be investigated and the reasons for studying it. If it was the effect of music on middle-aged adults (line 103), is there any theoretical reason why the authors suspect that the impact of music on this age group might be different from young adults, or is it simply because it has not been studied in this age group yet? The same applies to the lower limbs. Is there any theoretical reason that justifies analyzing the effect of music on the lower limbs, or is it simply because it has not been studied yet?
- Would self-selected music also be motivational? What is the difference between SSM and MM? Please, clarify in the introduction.

Experimental design

I think some criticisms and suggestions need to be addressed before the article is approved for publication. My major concern is about the characteristics of the songs in the SSM condition. There is no information about the Tempo of the songs used in this condition. Therefore, it is not possible to discriminate whether the better mean force performance observed in the SSM was due to Tempo or some other characteristic of the self-selected songs. By chance, all participants could have selected very stimulating songs, which could be a bias in the results. Did any participant choose a calm and relaxing song? I think that this information is essential for the reader of this article to understand the music influence on the performance of this type of exercise.
- Were the participants previously aware of the study's purpose? If so, consider including in the discussion/limitation how this may have interfered with the results.
- If my interpretation of the experimental design is correct, it could be that some self-selected music was also on the motivational playlist. Had this happened to any participant? If yes, how did the authors deal with this bias? If you present individual results (e.g., separated lines for each participant) in Figure 2, it would be possible to visualize individual interactions between MM and SSM songs.
- (line 149) – How was the MM tempo determined?
- Was there any control over the tempo (bpm) of the auto-selected songs?
- It would be interesting if the description of the test times were more precise. The same participant may have performed one test at 10 am and the other at 5 pm. If so, consider discussing this possible interference with the outcome. (DOI: 10.1519/JSC.0000000000003758; DOI: 10.1080/02701367.2020.1751032.)
- (line 165) – Considering that you used a modified equipment to acquire strength data, it would be interesting to include a real photo of the force data acquisition setup.
- I recommend you include more information about force acquisition and analysis (e.g., load cell acquisition frequency, signal filtering parameters, software used for data processing, etc.)

Validity of the findings

- I found the RPE very low for 12 maximal isometric reps. Perhaps the intensity of perceived exertion may explain some differences between your results and previous studies.
- Figure 3 C – suggestion to plot the values of the 12 repetitions. It would allow visualizing the effect of the music throughout the series of repetitions, including in relation to fatigue.

Discussion
- (line 276) – What was the Tempo (bpm) of SSM? In my opinion, it is not possible to make this statement without this information.

---

## Round 0.2 · accepted · Accept

The reviewers felt you adequately addressed their comments.

Reviewer 1 ·

Basic reporting

The authors have addressed all of the comments made for "Basic Reporting" during the intial review.

Experimental design

The authors have sufficiently addressed all of the comments made for "Experimental Design" during the intial review.

One lingering thought is that the revision added in line 142 regarding prevention of Hawthorne effects does not fully address the issue, but revision may not be necessary unless the authors wish to add more discussion in the limitations section of the paper. Hawthorne effects essentially refer to research participants behaving differently because they know that someone (i.e., a researcher), is observing them. They may choose songs differently solely on this basis. One way to try and account for this in experimental design would be to make the subjects aware that researchers would be absolutely blinded to the subjects' song choices, where maybe all playlist submissions were anonymous.

Validity of the findings

The authors have addressed all of the comments that I had regarding "Validity of the Findings".

Additional comments

With the revisions made to the current version of the manuscript, the authors have sufficiently addressed all of the concerns that I had during the first round of review of the original version. I believe that the manuscript is suitable for publication.

Reviewer 2 ·

Basic reporting

The authors revised the manuscript by taking into consideration the reviewers' comments.
Most of the previously presented criticisms have been solved. For this reason, I consider that the article meets the requirements to be accepted for publication.
Below are minor comments about the manuscript.

Experimental design

- (line 200) – It seems that the knee angle described does not match that one in Figure 1. Please check it.
– How was the MM tempo determined? I suggest inserting the author´s response to this question in the paper.

Validity of the findings

- After seeing your new Figure 4 I have a suggestion to you think and see if it's worth it. If you run a two-way ANOVA [condition (SSM x MM x CC) and repetition (1 up12)] maybe, you could see an interesting interaction. Perhaps the effect of SSM on mean force disappears as fatigue develops (e.g., after the 10th repetition). If this result is confirmed it would bring a relevant finding to the discussion.